# Binge-pattern alcohol consumption and genetic risk as determinants of alcohol-related liver disease

Chengyi Ding[1,14], Linda Ng Fat[2,14], Annie Britton[2], Pek Kei Im [3], Kuang Lin[3], Anya Topiwala [4], Liming Li[5,6,7], Zhengming Chen [3,8], Iona Y. Millwood [3,8], Steven Bell [9,10,15] ✉ & Gautam Mehta [11,12,13,15] ✉

Alcohol-related liver disease (ARLD) represents a major public health burden. Identification of high-risk individuals would allow efficient targeting of public health interventions. Here, we show significant interactions between pattern of drinking, genetic predisposition (polygenic risk score, PRS) and diabetes mellitus, and risk of incident ARLD, in 312,599 actively drinking adults in UK Biobank. Binge and heavy binge drinking significantly increase the risk of alcohol-related cirrhosis (ARC), with higher genetic predisposition further amplifying the risk. Further, we demonstrate a pronounced interaction between heavy binge drinking and high PRS, resulting in a relative excess risk due to interaction (RERI) of 6.07. Diabetes consistently elevates ARC risk across all drinking and PRS categories, and showed significant interaction with both binge patterns and genetic risk. Overall, we demonstrate synergistic effects of binge drinking, genetics, and diabetes on ARC, with potential to identify high-risk individuals for targeted interventions.

Alcohol-related liver disease (ARLD) is a major, and increasing, global health burden[1,2]. In the UK, ARLD deaths have increased by ~20% since the Covid-19 pandemic, and liver disease is forecast to become the most common cause of premature death within the next decade[3,4]. Similar trends are seen in the USA and across Asia[5,6]. In the UK, at least, this marked rise has occurred without an increase in population-level alcohol consumption, although changes in volume and/or pattern of alcohol use have been observed in heavy drinkers[3].

Alcohol use is the main risk factor for ARLD, and alcohol consumption is positively correlated with the risk of ARLD[7,8]. However, only about 1-in-3 heavy drinkers develop alcohol-related cirrhosis (the end stage of ARLD), and case-control and cohort studies demonstrate considerable heterogeneity in risk[8–13]. Other modifiers of risk have been investigated, including sex, genetic susceptibility, diabetes status, obesity, and environmental factors[14–16]. Genetic factors, in particular, have been of interest, since risk stratification on the basis of polygenic risk score (PRS) has the potential to identify high-risk

[1]Division of Psychiatry, University College London, London, UK. [2]Research Department of Epidemiology and Public Health, University College London, London, UK. [3]Clinical Trial Service Unit and Epidemiological Studies Unit (CTSU), Nuffield Department of Population Health, University of Oxford, Oxford, UK. [4]Big Data Institute, Nuffield Department of Population Health, University of Oxford, Oxford, UK. [5]Department of Epidemiology and Biostatistics, School of Public Health, Peking University Health Science Center, Beijing, China. [6]Key Laboratory of Epidemiology of Major Diseases (Peking University), Ministry of Education, Beijing, China. [7]Peking University Center for Public Health and Epidemic Preparedness & Response, Beijing, China. [8]Medical Research Council Population Health Research Unit (MRC PHRU), Nuffield Department of Population Health, University of Oxford, Oxford, UK. [9]Precision Breast Cancer Institute, Department of Oncology, University of Cambridge, Cambridge, UK. [10]Cancer Research UK Cambridge Centre, Li Ka Shing Centre, University of Cambridge, Cambridge, UK. [11]Institute for Liver and Digestive Health, University College London, London, UK. [12]Roger Williams Institute of Hepatology, Foundation for Liver Research, London, UK. [13]Royal Free London NHS Foundation Trust, London, UK. [14]These authors contributed equally: Chengyi Ding, Linda Ng Fat. [15]These authors jointly supervised this work: Steven Bell, Gautam Mehta. ✉e-mail: scb81@medschl.cam.ac.uk; gautam.mehta@ucl.ac.uk

individuals in whom to focus behavioral interventions or to consider for clinical trials[17].

Binge-pattern drinking, or heavy episodic alcohol intake, has generally been considered to be an independent risk factor for ARLD progression although human data to support this are scarce. Important experimental data from Bala et al. demonstrate that an acute ethanol binge increases circulating levels of bacterial proteins (lipopolysaccharide) and pro-inflammatory cytokines, both key in the pathogenesis of ARLD[18]. In terms of population-based studies, Aberg et al.[19], studying a well-characterized Finnish cohort, demonstrated that binge drinking episodes were significantly associated with liver-related hospitalization and death independent of average daily ethanol intake and age. More recently, Surial et al. found similar associations in a Swiss cohort with underlying HIV infection[20].

Moreover, identification of high-risk populations has been proposed as a strategy for stratified screening for liver fibrosis and delivery of targeted public health interventions[21]. Although drinking behavior and genetic factors have been studied as risk factors for ARLD, the specific interactions between pattern of drinking, genetic risk, and metabolic risk factors have not been previously studied. The specific aim of this study was to analyze data from the UK Biobank cohort, to test the hypothesis that both binge-pattern drinking and genetic risk factors were independently, and monotonically, associated with the risk of alcohol-related cirrhosis (ARC) and alcoholic hepatitis (AH). In this context, a monotonic relationship describes the relationship where variables change in the same direction. Additionally, we determined the degree to which pattern of alcohol use, genetic risk, and metabolic factors interact to influence ARC risk, since this may identify individuals for targeted behavioral intervention.

## Results

### UK Biobank participant characteristics

From the initial sample of 502,460 UK Biobank (UKB) participants, 6721 were excluded due to liver disease or viral hepatitis prior to baseline. Of the remaining participants, there were 342,541 current weekly drinkers. Data sets were also incomplete for a further 29,942 participants, predominantly due to missing physical activity (missing = 13,760), PRS (missing = 9481), and alcohol intake data (missing = 6177); the final UKB study cohort was 312,599 participants. Compared to the final study cohort, the participants with incomplete data had lower weekly alcohol consumption, were less likely to be binge or heavy binge drinkers but had higher ARC and AH incidence.

Those who drank within limits accounted for 20% of the sample, those who drank above limits, binge, or heavy binge accounted for 42%, 23%, and 15% respectively (Supplemental Table 1). Binge and heavy binge drinkers were younger than the average (mean age: binge group 68.7 years; heavy binge group 67.4 years vs. total sample 69.4 years), and more likely to be male (binge group 55%; heavy binge group 63% vs. total sample 51%). Additionally, binge and heavy binge drinkers drank less frequently in the week (e.g., 1–3 times a week: heavy binge 51% vs. total sample 37%), were more likely to drink without meals (e.g. heavy binge 33% vs. total sample 20%), more likely to drink beer (e.g., heavy binge 40% vs. total sample 30%), and more likely to be current smokers (e.g., heavy binge 18% vs. total sample 10%).

Characteristics of the UKB study cohort, separated by disease phenotype, are presented in Table 1. The cohort comprised 734 cases of ARC, and 136 cases of AH over a median follow-up of 12.6 years (interquartile range 11.9, 13.3 years). Liver disease cases were significantly more likely (by post-hoc pairwise comparison with Bonferroni correction) to be male (ARC 78% vs. control 51%, $p < 0.001$; AH 77% vs. control 51%, $p < 0.001$), current smokers (ARC 33% vs. control 10%, $p < 0.001$; AH 32% vs. control 10%, $p < 0.001$) and to be less active (no physical activity: ARC 53% vs. control 35%, $p < 0.001$; AH 50% vs. control 35%, $p = 0.027$). As expected, greater weekly alcohol intake and daily drinking were significantly associated with risk of liver disease (mean weekly alcohol intake: ARC 502 g vs. control 186 g, $p < 0.001$; AH 577 g vs. control 186 g, $p < 0.001$; daily or almost daily drinking: ARC 71% vs. control 30%, $p < 0.001$; AH 71% vs. control 30%, $p < 0.001$).

**Table 1 | Baseline characteristics of 312,599 current weekly drinkers in the UK Biobank study by disease status**

| | ARLD-free controls (n = 311,817) | ARC (n = 734) | AH (n = 136) |
|---|---|---|---|
| Age, years[a] | 69.4 (8.0) | 70.1 (7.7) | 67.3 (8.0) |
| Weekly alcohol intake, g per week[a] | 186.4 (158.4) | 502.4 (392.0) | 576.8 (413.6) |
| Alcohol consumption group[b] | | | |
| Within daily limit | 61,694 (19.8) | 47 (6.4) | 4 (2.9) |
| Above daily limit but below binge | 131,851 (42.3) | 151 (20.6) | 23 (16.9) |
| Binge | 71,320 (22.9) | 194 (26.4) | 36 (26.5) |
| Heavy binge | 46,952 (15.1) | 342 (46.6) | 73 (53.7) |
| Male | 159,222 (51.1) | 570 (77.7) | 105 (77.2) |
| Ethnicity, white | 294,086 (94.3) | 684 (93.2) | 127 (93.4) |
| Smoking status | | | |
| Never | 162,465 (52.1) | 208 (28.3) | 46 (33.8) |
| Ex-smoker | 118,765 (38.1) | 284 (38.7) | 46 (33.8) |
| Current smoker | 30,587 (9.8) | 242 (33.0) | 44 (32.4) |
| Physical activity[c] | | | |
| None | 107,730 (34.5) | 390 (53.1) | 68 (50.0) |
| 1–2 days | 98,747 (31.7) | 140 (19.1) | 25 (18.4) |
| 3+ days | 105,340 (33.8) | 204 (27.8) | 43 (31.6) |
| Frequency of drinking | | | |
| Daily or almost daily | 91,981 (29.5) | 519 (70.7) | 97 (71.3) |
| 3–4 times a week | 105,024 (33.7) | 136 (18.5) | 29 (21.3) |
| 1–3 times a week | 114,812 (36.8) | 79 (10.8) | 10 (7.4) |
| Alcohol type | | | |
| Mixed | 161,529 (51.8) | 350 (47.7) | 61 (44.9) |
| Wine only | 8494 (2.7) | 3 (0.4) | 0 (0.0) |
| Beer only | 92,994 (29.8) | 276 (37.6) | 53 (39.0) |
| Spirits only | 48,800 (15.7) | 105 (14.3) | 22 (16.2) |
| Drinking with meals | | | |
| Not with meals | 63,332 (20.3) | 293 (39.9) | 61 (44.9) |
| With meals | 136,642 (43.8) | 149 (20.3) | 20 (14.7) |
| It varies | 111,843 (35.9) | 292 (39.8) | 55 (40.4) |
| PRS score | | | |
| Low | 63,469 (20.4) | 71 (9.7) | 20 (14.7) |
| Middle | 188,125 (60.3) | 373 (50.8) | 77 (56.6) |
| High | 60,223 (19.3) | 290 (39.5) | 39 (28.7) |
| BMI, kg/m² | | | |
| <18.5 | 1357 (0.4) | 10 (1.4) | 1 (0.7) |
| 18.5–24.9 | 107,780 (34.6) | 192 (26.2) | 38 (27.9) |
| 25–29.9 | 138,124 (44.3) | 257 (35.0) | 54 (39.7) |
| ≥30 | 64,556 (20.7) | 275 (37.5) | 43 (31.6) |
| Prevalent diabetes | 11,738 (3.8) | 103 (14.0) | 6 (4.4) |

Values are numbers (percentages) unless otherwise indicated.
[a]Mean (standard deviation).
[b]Based on average daily alcohol consumption: within daily limit (<24 g for women, <32 g for men), above daily limit but below binge (24–48 g for women, 32–64 g for men), binge (48–72 g for women, 64–96 g for men) and heavy binge (≥72 g for women, ≥96 g for men).
[c]Physical activity based on the number of days engaged in vigorous activity for 10 min or more.

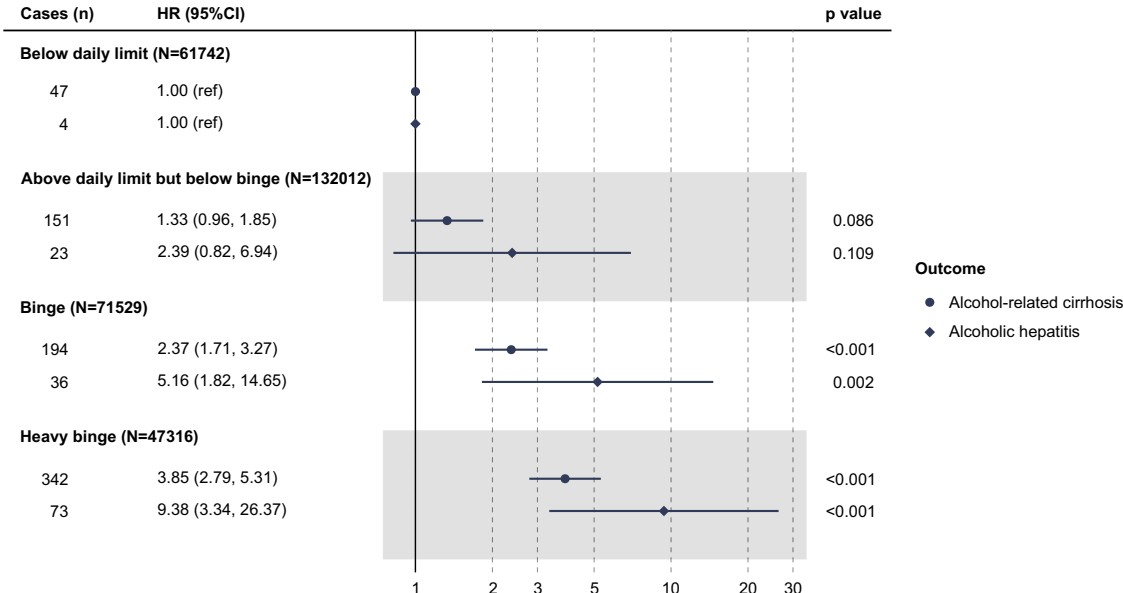

**Fig. 1 | Risk of advanced ARLD associated with alcohol consumption group.** Data are presented as hazard ratios (HRs) with respective 95% confidence intervals (CIs). All the *p* values were two-sided and calculated using the Wald test. HRs were derived from Cox model with the following covariates: alcohol consumption group, sex, age, ethnicity, Townsend deprivation index at recruitment, physical activity, smoking, total weekly alcohol intake, beverage type, drinking with/without meal, prevalent diabetes, and BMI categories.

## Binge drinking and ARC and AH

In models based on observational data only without adjusting for polygenic risk score (PRS) (Fig. 1), there was an elevated risk of ARC with greater daily consumption (*p* for trend <0.001); for those drinking above daily limits but below binge levels the hazard ratio (HR) was 1.33 (95% confidence interval, 95% CI 0.96, 1.85), which increased to levels of statistical significance for binge-pattern drinking (HR 2.37; 95% CI 1.71, 3.27) and heavy binge drinkers (HR 3.85; 95% CI 2.79, 5.31). Similarly, for AH, risk increased with greater daily consumption. There was a trend of increased risk in subjects with increasing alcohol consumption, including binge-pattern drinking (HR 5.16; 95% CI 1.82, 14.65) and heavy binge drinking (HR 9.38; 95% CI 3.34, 26.37; *p* for trend <0.001). Importantly, these models were adjusted for total weekly alcohol intake, as well as sex, age, prevalent diabetes mellitus, BMI, and other socio-demographic and behavioral risk factors. Moreover, no interaction was found between alcohol consumption and sex or continuous age (*p* for interaction ranged from 0.17 to 0.49), suggesting that the effect of binge-pattern drinking on ARC and AH is not significantly different between men and women and across different ages.

## Polygenic risk and ARC and AH

After accounting for multiple covariates including binge drinking pattern, weekly alcohol intake, diabetes mellitus, and obesity, the risk of developing ARC increased monotonically with increasing PRS (Supplemental Table 2; all *p* for trend <0.001). For example, compared with the low PRS group, the HR of ARC for the middle and high PRS groups was 1.60 (95% CI 1.24, 2.07) and 3.51 (95% CI 2.70, 4.55), respectively. A less steep gradient was seen for AH, although the association did not reach statistical significance, likely due to the smaller number of cases in each subgroup. Again, no significant interaction was observed between PRS group and sex or continuous age (*p* for interaction ranged from 0.26 to 0.94). Higher PRS was also associated with a higher risk of advanced ARC and AH when modeled as a standardized continuous variable (Supplemental Table 2).

## Interaction of binge drinking with polygenic risk on ARC

Analyses of interactions between pattern of alcohol consumption and genetic risk were restricted to the primary endpoint of ARC-related hospitalization or death, since the number of AH cases was relatively small and no significant associations of categorical PRS score and AH were found (above and Supplemental Table 2). Within each alcohol consumption group, increasing PRS led to an increased risk of ARC (Fig. 2). This was most marked in the heavy binge group, where PRS influenced risk of ARC from HR 3.53 (95% CI 1.35, 9.19) in the lowest PRS group to HR 12.82 (95% CI 5.21, 31.52) in the highest PRS group. Again, importantly, these associations were independent of total weekly alcohol intake, as well as sex, age, prevalent diabetes mellitus, body mass index (BMI), and other sociodemographic and behavioral risk factors.

We found no evidence for a multiplicative interaction between alcohol consumption group and PRS group (*p* = 0.689 for likelihood ratio test comparing models with and without interaction terms). However, there was evidence of an additive interaction between heavy binge and PRS group (Supplemental Table 3). Specifically, the relative excess risk due to interaction (RERI) was 6.07 (95% CI 0.20, 11.94) for the presence of heavy binge and high PRS.

## Interaction of diabetes mellitus with binge drinking and polygenic risk on ARC

Prior studies have shown diabetes is an independent risk factor for cirrhosis[22]. Additionally, recent work suggested that inclusion of diabetes status enhanced the utility of a three single-nucleotide polymorphism (SNP) score for ARC, although no specific multiplicative interaction was found[17]. Consequently, we analyzed interaction effects of diabetes status with binge drinking and PRS on ARC in the cohort presented here. Further, we extended our model to perform exploratory analyses of three-way interaction between these factors.

Diabetes status was found to significantly increase risk of ARC across all alcohol consumption groups, and also in all three categories of PRS, after adjustment for sex, age, BMI, and other relevant risk factors (Fig. 3). No multiplicative interaction was observed between diabetes and binge drinking or diabetes and PRS group on risk of ARC (*p* for likelihood ratio test = 0.99 and 0.92, respectively). Diabetes, however, had significant two-way additive interactions with heavy binge drinking and high PRS, with a RERI of 4.69 (95% CI 0.50, 8.88) and 4.83 (95% CI 0.99, 8.67), respectively (Supplemental Table 4).

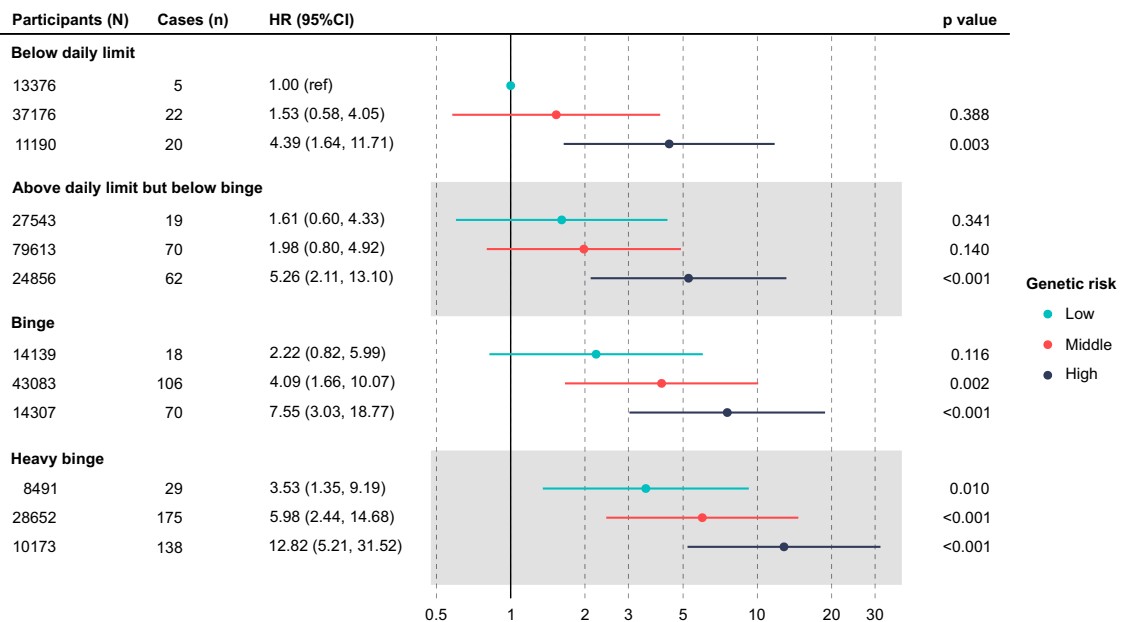

**Fig. 2 | Risk of alcohol-related cirrhosis by alcohol consumption and polygenic risk group.** Data are presented as hazard ratios (HRs) with respective 95% confidence intervals (CIs). All the *p* values were two-sided and calculated using the Wald test. HRs were derived from Cox model with following covariates: interaction of alcohol consumption and PRS groups, total weekly alcohol intake, diabetes, BMI, sex, age, ethnicity, Townsend deprivation index, physical activity, smoke, alcohol type, drinking with/without meals, genotyping array, first 10 ancestry principal components.

Finally, we proceeded to examine whether three-way interactions between these three factors (heavy binge drinking, high PRS, diabetes) were present. As shown in Fig. 4, the excess risk due to the joint presence of heavy binge, high PRS, and diabetes compared to each risk factor acting individually was substantial, with a total RERI of 24.39 (95% CI −0.24, 49.02). The corresponding AP and S were 0.85 (95% CI 0.73, 0.97) and 8.12 (95% CI −0.67, 16.91), respectively. The RERI owing to three-way interaction beyond two-way interactions was −3.63 (95% CI −26.73, 19.46), indicating that there was no excess risk which is explicitly due to the three-way interaction between heavy binge, high PRS, and diabetes.

### Sensitivity analyses and external validation

Results and interpretation did not change when we accounted for the potential competing risk of death from non-ARLD causes (Supplemental Tables 5–8) or when we used multiply imputed data sets (Supplemental Tables 9–13).

Analyses of binge-pattern drinking and risk of ARC and AH were conducted in male weekly drinkers within an external cohort of CKB (N = 69,039), with baseline characteristic of these participants presented in Supplemental Tables 14a and 14b. These models also demonstrate that binge-pattern drinking was associated with higher risk of ARC and AH, independent of total weekly alcohol intake (Supplemental Table 15). Within each alcohol group, diabetes conferred a higher risk of ARC (Supplemental Table 16); note that the two lowest consumption groups were combined due to zero cases of ARC in participants who reported drinking below daily limit and had diabetes. Additionally, diabetes had an additive interaction with heavy binge drinking, with a RERI of 4.37 (95% CI −1.14, 9.88; Supplemental Table 17).

In the China Kadoorie Biobank (CKB) over 30,760 males with PRS data, after excluding prior liver cirrhosis or hepatitis, were included in preliminary analyses of genetic risk of ARC and AH. One SD increase in PRS was directionally consistent but not significantly associated with ARC (HR 1.13; 95% CI 0.86, 1.48) and AH (HR 1.13; 95% CI 0.65, 1.97) in genotyped men, or among male weekly drinkers (ARC HR 1.08; 95% CI 0.79, 1.48; AH HR 0.80; 95% CI 0.42, 1.52), possibly due to a lack of

statistical power (Supplemental Table 18). Consequently, further analyses of interaction between PRS and binge-pattern/diabetes in this population were not pursued.

Finally, validation was performed in random UKB sub-samples, which also supported our initial findings, albeit with wider CIs due to reduced sample sizes (Supplemental Tables 19–23). Although there were some differences between the point HR estimates in the main analyses and in CKB or the UKB sub-samples for certain strata with fewer ARC/AH cases, the CIs often overlapped and included the point estimates.

### Discussion

This large, population-based study demonstrates a synergistic interaction between binge-pattern drinking and genetic factors in the development of ARC. Specifically, these data demonstrate an independent effect of binge-pattern alcohol consumption on risk of ARC and AH. Further, within categories of binge drinking or heavy binge drinking, any increase in genetic risk score was associated with an elevated risk of developing ARC. Moreover, the interaction between these factors indicates a ~sixfold increase in risk when both heavy binge drinking and genetic risk are present, compared to each factor acting independently. Finally, diabetes mellitus was also noted to independently influence risk of ARC due to heavy binge drinking and genetic susceptibility, with significant two-way additive interactions found. These data have translational applicability, through the potential risk-stratification of drinkers based on pattern of drinking, genetic risk, and history of diabetes mellitus. In particular, if these data are further validated in additional cohorts, they provide the basis for development of a risk score for targeting behavioral interventions or screening strategies for ARLD.

Few studies have addressed pattern of drinking as an independent risk factor for advanced ARLD. Aberg et al. demonstrated an independent effect of binge-pattern drinking in the Finnish Heath 2000 study comprising just over 6000 participants, with a hazard ratio between 3.0–4.5 for weekly binge drinking with liver-related events (liver-related hospitalization, liver cancer, or liver-related death)[19]. Surial and colleagues found binge-pattern drinking was associated

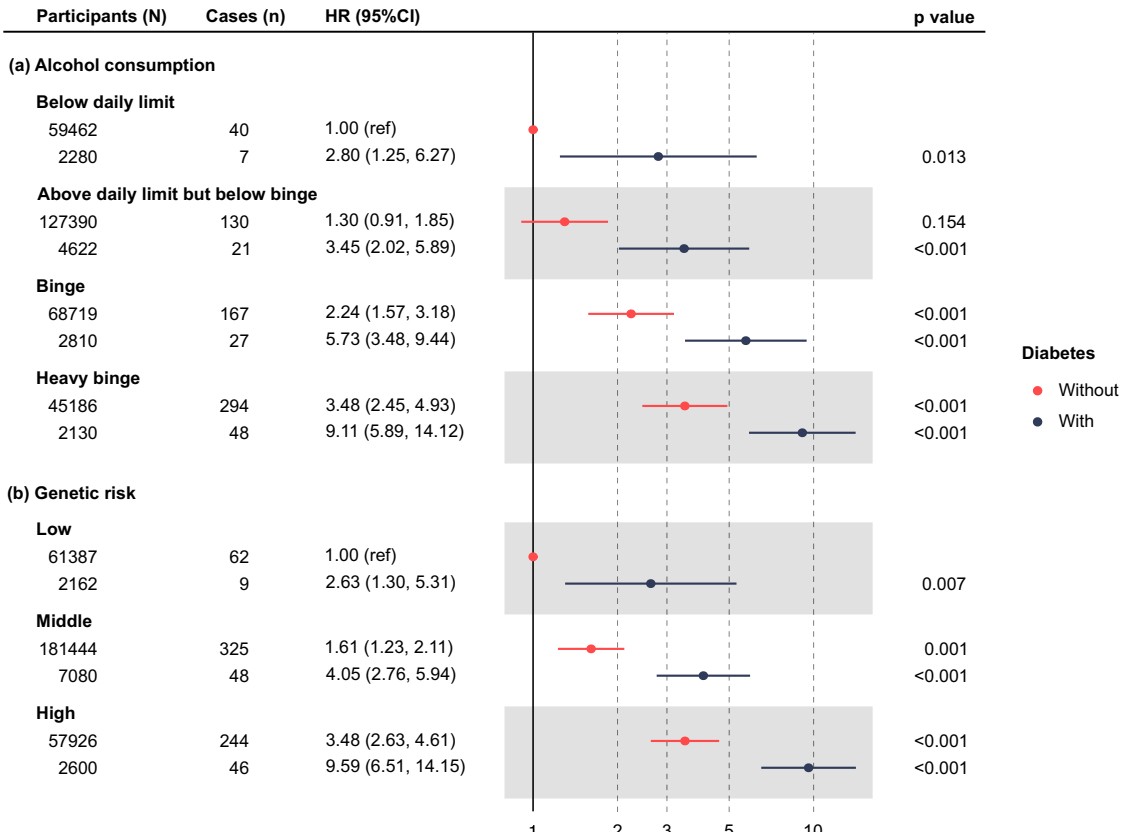

| Participants (N) | Cases (n) | HR (95%CI) | p value |
|---|---|---|---|
| **(a) Alcohol consumption** | | | |
| **Below daily limit** | | | |
| 59462 | 40 | 1.00 (ref) | |
| 2280 | 7 | 2.80 (1.25, 6.27) | 0.013 |
| **Above daily limit but below binge** | | | |
| 127390 | 130 | 1.30 (0.91, 1.85) | 0.154 |
| 4622 | 21 | 3.45 (2.02, 5.89) | <0.001 |
| **Binge** | | | |
| 68719 | 167 | 2.24 (1.57, 3.18) | <0.001 |
| 2810 | 27 | 5.73 (3.48, 9.44) | <0.001 |
| **Heavy binge** | | | |
| 45186 | 294 | 3.48 (2.45, 4.93) | <0.001 |
| 2130 | 48 | 9.11 (5.89, 14.12) | <0.001 |
| **(b) Genetic risk** | | | |
| **Low** | | | |
| 61387 | 62 | 1.00 (ref) | |
| 2162 | 9 | 2.63 (1.30, 5.31) | 0.007 |
| **Middle** | | | |
| 181444 | 325 | 1.61 (1.23, 2.11) | 0.001 |
| 7080 | 48 | 4.05 (2.76, 5.94) | <0.001 |
| **High** | | | |
| 57926 | 244 | 3.48 (2.63, 4.61) | <0.001 |
| 2600 | 46 | 9.59 (6.51, 14.15) | <0.001 |

Diabetes
● Without
● With

**Fig. 3 | Risk of alcohol-related cirrhosis by diabetes status and both alcohol consumption and polygenic risk group.** Data are presented as hazard ratios (HRs) with respective 95% confidence intervals (CIs). All the *p* values were two-sided and calculated using the Wald test. HRs were derived from Cox models with the following covariates: sex, age, ethnicity, Townsend deprivation index at recruitment, physical activity, smoking, total weekly alcohol intake, beverage type, drinking with/without meal, BMI, genotyping array, first 10 ancestry principal components plus PRS group, the interaction of alcohol consumption group and diabetes for (**a**), and alcohol consumption group, interaction of PRS group and diabetes for (**b**). Data are presented as HR and 95% confidence interval.

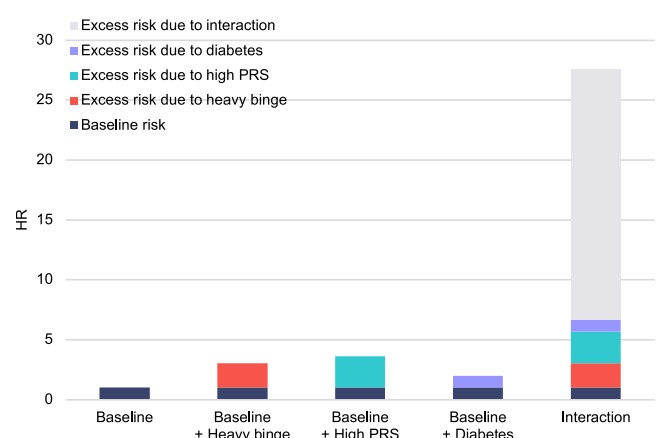

**Fig. 4 | Hazard ratios for risk of alcohol-related cirrhosis with contributions of heavy binge, high PRS, diabetes, and their combination (interaction).** Baseline represents the risk in referent participants who reported drinking within daily limits, had low PRS, and were free of diabetes. Source data are provided as a Source Data file.

with liver-related events in a cohort of HIV-positive patients with a similar degree of elevation in risk[20]. The data presented here extend this previous work in a larger cohort with more liver-related events, and also the demonstration of a graded increase in binge drinking (heavy binge; women: >72 g/day, men: >96 g/day) conferring a much higher degree of risk for ARC (~4-fold).

Further, these data demonstrate a similar graded increase in risk of AH with degree of binge alcohol consumption. Although binge drinking is widely considered to be a major risk factor for AH, a severe and fulminant form of ARLD, there are few data to support this. In a French cohort, Naveau et al. showed that female sex and obesity were risk factors for AH, although volume of alcohol consumption was not significantly related[23]. More recently, in a case-control study, Liang-punsakul et al. noted that AH patients consumed less alcohol with lower levels of binge drinking; the authors suggested genetic susceptibility to be an overarching factor[24]. By contrast, our findings support a graded increase in risk of AH with binge-pattern drinking, which is consistent with the prevalent view in ARLD guidance documents.

Genetic factors are well known to contribute to risk of both ARC and AH. However, to our knowledge, this is the first study to demonstrate an interaction of genetic risk with pattern of drinking in the development of advanced ARLD. Whitfield et al. recently demonstrated that a 3 SNP-based risk score distinguished ARC from controls in an at-risk population drinking >80 g/day (men) or >50 g/day (women) with an odds ratio of 2.7–5.0 in the highest tertile of risk[17]. However, no specific assessment of pattern of drinking was made. When diabetes status was added, the risk of ARC increased to HR 14.7–17.1 for the highest tertile of genetic risk, although no specific multiplicative interaction was found. Our findings augment these data by demonstrating a monotonic increase in risk of ARC with increasing PRS, an additive interaction between binge drinking and PRS on ARC, and, importantly, significant two-way additive interactions between diabetes and PRS, and diabetes and heavy binge drinking. Thus, these

data also extend previous observations by demonstrating a synergistic interaction between diabetes mellitus and genetic factors, and diabetes mellitus and binge-pattern drinking, on the risk of cirrhosis.

The results of this study should be interpreted in light of some of the limitations. Firstly, although the majority of our findings were validated in the external CKB cohort, we were unable to replicate our findings demonstrating an association of PRS with ARC in this cohort. This is likely a consequence of insufficient statistical power, although it may also reflect decreased utility of the PRS in non-European populations. Further validation is required to address this, to develop a model for disease risk. Validation in non-European populations would also be important for broader applicability. Secondly, our cohort was also insufficiently powered for three-way interactions and, consequently, these analyses must be considered exploratory, although two additive interactions were replicated and we believe are robust. Thirdly, the UK Biobank represents a relatively healthy and affluent cohort, potentially explaining the relatively low incidence of ARC- and AH-related hospital admissions. Nevertheless, this ARLD cohort is larger than most reported cohorts in whom detailed alcohol use data are available[25]. Fourth, a difference in characteristics was noted between the final UKB study cohort and those in whom data sets were incomplete, with higher ARLD incidence in the participants with missing data despite seemingly lower alcohol binge-pattern alcohol consumption. This observation may be consistent with the 'sick quitter hypothesis', whereby participants may reduce alcohol consumption due to ill health[26]. Finally, the calculation of binge and heavy binge categories was based on self-reported weekly intake and frequency of drinking, and consequently may have been imprecise although, in turn, would be likely to underestimate any interaction of binge or heavy binge drinking with ARLD. Additionally, data regarding pattern of drinking was only collected at baseline.

In conclusion, these data demonstrate robust associations of binge-pattern drinking, genetic susceptibility, and diabetes mellitus with risk of ARC in a UK-based population cohort, and novel, synergistic interactions between these risk factors. Additionally, interactions between binge-pattern drinking and diabetes mellitus were validated in an external, Chinese cohort. These findings support stratification of drinkers for cirrhosis screening, behavioral intervention, or selection for clinical trials, based on a combination of these risk factors. Additional validation and prospective testing of clinical risk score based on pattern of drinking and/or genetic risk is warranted given the heavy and rising burden of ARLD worldwide.

# Methods

## Study and participants

This study uses data from the UK Biobank, a large prospective population-based study of over 500,000 adults aged 40–69 years. UK Biobank has approval from the UK North West Multi-center Research Ethics Committee (MREC, reference 16/NW/0274) as a Research Tissue Bank (RTB). All participants provided written, informed consent. Data were collected between 2005 and 2010 from 22 UK Biobank Assessment Centers. Invitations were sent to individuals if they lived within 25 miles of an assessment center and were registered with a general practitioner. Consent for data linkage was provided by participants. Further information on the study can be found elsewhere[27]. Participants with any liver disease or viral hepatitis before baseline were excluded. Additionally, this sample was limited to current weekly drinkers, due to potential bias from health selection effects of recently stopping drinking (for example due to ill health), as done elsewhere[28]. Further, participants with incomplete data for all variables were also excluded.

We validated our findings in an external cohort, the China Kadoorie Biobank (CKB)[29]. CKB is a prospective cohort study of 512,724 participants, aged 30–79 years at enrollment, from 5 urban and 5 rural areas in China. Ethical approval was obtained from the Ethical Review

Committee of the Chinese Center for Disease Control and Prevention (Beijing, China) and the Oxford Tropical Research Ethics Committee, University of Oxford (UK). Informed consent was obtained from all participants included in the study. Baseline assessment took place in 2004–2008 and participants with prior liver cirrhosis or hepatitis were excluded. Analyses in CKB were restricted to male weekly drinkers as only 2% of female participants reported regular drinking.

## Alcohol consumption

At the initial recruitment, participants were asked to provide detailed information on alcohol consumption via a touchscreen questionnaire. Participants were asked how often they drank alcohol; responses included: daily or almost daily, 3–4 times a week, 1–2 times a week, monthly or less (not included). Those who drank alcohol at least weekly were asked to report how many glasses they drank in an average week for each beverage type. A glass was converted into alcohol units based on standard conversion: beer 2.5 units, white wine 2.1, fortified wine 1, red wine 2.1, spirits 1, other alcoholic drinks 1.5, and subsequently converted into grams (g) of ethanol (one unit is equivalent to 8 g of ethanol).

Average daily consumption was calculated by dividing the total weekly grams by the mid-point of categorical response to the frequency question above (daily or almost daily, 3–4 times a week, 1–2 times a week) by 6, 3.5 and 1.5 respectively, to estimate an average daily amount. A categorical variable was created based on average daily alcohol consumption and standard cut-offs and definitions for daily and binge consumption used in the UK the following way: within daily limits (<24 g/day for women, <32 g/day for men (reference)), above daily limits but below binge (≥24 g and <48 g/day for women, ≥32 g and <64 g/day for men), binge (≥48 g and <72 g/day units for women, ≥64 g and <96 g/day units for men) and heavy binge (≥72 g/day for women, and ≥96 g/day for men).

Detailed information was collected on the type of beverages that participants drank in the previous week. This was classified into wine-only drinkers, beer only drinkers, spirit only drinkers and those who drank a mixture of beverages. A question on drinking with meals was asked with responses including: drinking with meals, not drinking with meals, and it varies.

## Liver-related endpoints

The primary objective was to investigate the relationship between pattern of alcohol consumption, genetic factors, and risk of alcohol-related cirrhosis (ARC; ICD-10 codes K70.2, K70.3, and K70.9). A secondary objective was to similarly investigate associations between pattern of alcohol consumption, genetic factors, and risk of alcoholic hepatitis (AH; K70.1). Importantly, less advanced stages of ARLD, such as alcoholic fatty liver (K70.0), were excluded. Baseline data was linked to hospital episode statistics and national mortality register, which are provided in the UK Biobank dataset. The primary endpoint event for this study was first liver-related hospitalization or death due to ARC. Secondary outcomes included first hospitalization or death due to AH.

## Polygenic risk score

We searched the PRS Catalog for alcoholic liver cirrhosis and selected the score with the best performance indicator at the time (May 2022)[30]. This was a PRS based on a weighted combination of multiple biomarker polygenic scores using 183,271 genetic variants[31]. The PRS was calculated as the linear combination of the beta (i.e. weight) coefficient multiplied by the number of effect alleles an individual carries for each genetic variant using PLINK2. Further details of the PRS are available at: www.pgscatalog.org/score/PGS000704 (risk score ID: PGS000704). The PRS was normally distributed across participants (Supplemental Fig. 1) and grouped according to quintiles as low risk (quintile 1), middle risk (quintiles 2, 3, and 4), and high risk (quintile 5).

## Covariates

Diabetes mellitus and obesity have been shown to contribute to risk of ARLD[17,32], and consequently both these factors were adjusted for in our models. Diabetes was physician-diagnosed, at baseline, and reported as a binary variable. Body mass index (BMI) was also calculated at baseline and categorized into <18.5, 18.5–24.9, 25–29.9, and ≥30 kg/m². Other covariates adjusted for included sex, age, ethnicity (white vs. non-white), area-deprivation based on a Townsend score—a standard measure of area deprivation that was assigned based on participants' residence and included as a continuous variable. Self-reported smoking status (never smokers/ex-smokers/current smokers) and physical activity based on the number of days engaged in vigorous activity for 10 min or more categorized into approximately equal groups (0/1–2/3–7 days).

## Statistical analyses

Initial descriptive statistics present characteristics of average daily alcohol consumption groups by all variables considered in the model. Chi-squared tests were conducted to check for bi-variate associations. The associations between measures of alcohol consumption and liver diseases were assessed using Cox Proportional Hazard models, with each liver disease included in separate models. The proportional hazard assumptions were tested with Schoenfeld residuals and were found not to be violated. Covariates were checked for multicollinearity using the variance inflation factor (VIF), but no multicollinearity was noted (all VIF ≤ 2.08). Survival time was calculated from the date of the baseline assessment to the date of the first occurrence of ARLD or death or censoring (up to the last date of data linkage 30th September 2021). We assessed interaction on both a multiplicative and additive scale as the presence or absence of interaction on either can be informative[33]. Additive interaction implies that the combined effect of two exposures is greater (or smaller) than the sum of the individual effects of the two exposures (and others have argued this reflects mechanistic interaction, rather than merely statistical interaction), whereas interaction on a multiplicative scale indicates that the combined effect is larger (or lesser) than the product of the individual effects. Additive interaction was assessed using three measures: RERI, the relative excess risk due to interaction; AP, the attributable proportion due to interaction; and S, the synergy index. If there is no additive interaction, RERI and AP are equal to 0 and S is equal to 1.

In sensitivity/additional analyses, we considered non-ARLD death as a competing risk using the Fine and Gray method[34]. To address missing information, we employed multiple imputation by chained equations[35]. ARLD was represented by an indicator variable, and the Nelson-Aalen cumulative hazard estimator was utilized[36]. In addition to an external validation using CKB, we also performed a cross-validation by randomly splitting our UKB sample into two independent sub-samples (evenly balanced by age and sex) and then repeated all the analyses in each sub-sample.

## Reporting summary

Further information on research design is available in the Nature Portfolio Reporting Summary linked to this article.

## Data availability

All data provided by the UKB are available to other investigators online upon permission granted by www.ukbiobank.ac.uk. Restrictions apply to the availability of these data, which were used under license for the current study (application ID 42520). The individual-level data on PRS and ARLD-related hospitalization generated in this study have been deposited in the UKB repository under upload ID 4475 and can be accessed with permission from the UKB. Similarly, data from CKB are available to open-access users upon permission granted from www.ckbiobank.org. A research proposal will be requested to ensure that any analysis is performed by bona fide researchers and, where data are not currently available to open-access researchers, is restricted to the topic covered in this paper. Source data are provided with this paper.

## Code availability

The code used for data analyses is available at: https://github.com/ChengyiDing/Binge_prs_UKBstudy.

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

## Acknowledgements

This work was supported by grant funding: the CKB baseline survey and the first re-survey were supported by the Kadoorie Charitable Foundation in Hong Kong. The long-term follow-up of the CKB study has been supported by Wellcome Trust grants to Oxford University (212946/Z/18/Z, 202922/Z/16/Z, 104085/Z/14/Z, 088158/Z/09/Z) and grants from the National Natural Science Foundation of China (82192900, 82192901, 82192904) and from the National Key Research and Development Program of China (2016YFC0900500). DNA extraction and genotyping was supported by grants from GlaxoSmithKline and the UK Medical Research Council (MC-PC-13049, MC-PC-14135). The UK Medical Research Council (MC_UU_00017/1, MC_UU_12026/2, MC_U137686851), Cancer Research UK (C16077/A29186; C500/A16896) and the British Heart Foundation (CH/1996001/9454), provide core funding to the Clinical Trial Service Unit and Epidemiological Studies Unit at Oxford University for the project. PKI is supported by an Intermediate Research Fellowship from the Nuffield Department of Population Health, University of Oxford. AT is supported by a Fellowship from the Wellcome Trust (216462/Z/19/Z). AB and GM are part-supported by NIHR funding (PB-PG-0418-20038). SB is supported by Cancer Research UK funding (A27657). For the purpose of Open Access, the authors have applied a CC-BY public copyright licence to any Author Accepted Manuscript version arising from this submission. We would also like to acknowledge the generous and invaluable contributions of the participants of the UKB and CKB studies.

## Author contributions

Study concept and design: L.N.F., C.D., A.B., S.B., G.M.; acquisition of data: L.N.F., C.D., A.B., P.K.I., K.L., A.T., L.L., Z.C., I.M., S.B., G.M.; analysis and interpretation of data: L.N.F., C.D., A.B., P.K.I., A.T., L.L., Z.C., I.M., S.B., G.M.; drafting of the manuscript: L.N.F., C.D., A.B., S.B., G.M.; critical revision of the manuscript for important intellectual content: L.N.F., C.D., A.B., P.K.I., A.T., L.L., Z.C., I.M., S.B., G.M.; statistical analysis: L.N.F., C.D., S.B.; study supervision: S.B., G.M.

## Competing interests

G.M. is an inventor of 'Treatment of Pyroptosis in Liver Disease' (Patent filing: US20210069296A1; EP19721333.3A), and is a co-founder of Hepyx Limited. The remaining authors declare no competing interests.
