## [Peer Review File · Nature Communications]

Binge-Pattern Alcohol Consumption and Genetic Risk as Determinants of Alcohol-Related Liver DiseaseREVIEWER COMMENTS

Reviewer #1 (Remarks to the Author):

This study aimed to determine the interactions between known genetic and metabolic factors and patterns of drinking in the UK Biobank cohort of active drinkers for identifying patients at risk of ARLD and AH. The researchers used Cox proportional hazards for association between self-reported patterns and amounts of alcohol use and incident liver disease, a predetermined polygenic risk score (PRS, using monotonic logic for genetic risk) and diabetes status. Additive and multiplicative interactions modelling for cirrhosis risk determined interactions between PRS, diabetes and drinking patterns. The results show increasing risk of cirrhosis with increasing amounts of alcohol use: heavy binge >binge vs within daily drinking limits; increased cirrhosis risk with higher PRS; presence of diabetes increased risk in all drinking categories as well as all PRS risk quintiles; additive interactions between: heavy binge and high PRS, diabetes and heavy binge, and diabetes and high PRS. The authors conclude that synergistic interactions between high binge, high PRS and diabetes significantly elevate the risk for alcohol-related cirrhosis.

The study methodology, quality of data, robustness of results and their interpretation are valid. The manuscript clearly define the hypothesis leading to aims, appropriate methodology and interpretation of results.

The significance of this research is high in the field of alcohol-related liver disease as so far there is no comprehensive risk assessment algorithm that can truly identify patients at risk. The robust interaction between genetic risk, and presence of diabetes with pattern of drinking (daily drinking vs binge vs heavy binge) is new information. This study adds to the current knowledge that is very much relevant clinically.

The manuscript can be improved with addition of further information on the PRS, such as number of SNPs included and perhaps even their list in supplemental data. What is the algorithm the authors used for calculating PRS. What are the cut-offs for low/medium/high PRS and how was it decided? It was difficult to get precise information for this study from the reference provided.

UK Biobank cohort, although large per se, is control heavy with very few cases especially for AH in this study which may affect the results. Discuss UK Biobank dataset which is control heavy compared to cases (both ARLD and AH) and what implications that might have on the results/interpretation. Drinking information is based on self-report and subject to bias when replicating the study.

Sharing a proposed risk algorithm for interactions based on their data will be useful for testing in other cohorts for furthering this field.

Results, para 1: The sentence below is not discussed in the manuscript. Can the authors speculate on the high incidence in these patients

“Compared to the final study cohort, the participants with incomplete data had lower weekly alcohol consumption, were less likely to be binge or heavy binge drinkers but had higher ARC and AH incidence.”

Minor

Check for typos throughout the manuscript

A bit more definition of ‘monotonic’ in the introduction will be helpful for readers non-conversant with this terminology

Reviewer #2 (Remarks to the Author):

This is a well-performed analysis from the large UK Biobank dataset. The authors found a significant additive interaction effect between heavy binge drinking and high genetic risk for alcohol-related cirrhosis. A similar additive interaction effect was found between diabetes and heavy binge drinking and between diabetes and high genetic risk for alcohol-related cirrhosis. These findings support the observations from previous studies, which have found considerable interaction effects between various levels and types of alcohol use and metabolic and/or genetic factors for liver disease. The current study is an important piece of work that adds on previous work and extend the analyses in several ways. However, I suggest toning down the statements of novelty (“...demonstrate, for the first time,...” etc).

Some issues deserve clarification:

- The mean age was quite high, around 70 years. Were the findings similar in younger individuals? Were there differences between men and women in the findings?
- There are many approaches to estimating total alcohol intake in the UK Biobank data. Has the approach that the authors used been validated or used previously?
- Is there an issue with multicollinearity in the models that seem to include both drinking pattern (binge) and total weekly alcohol intake? How did you assess and treat such collinearity?
- What about the competing risk of death without liver disease? An additional analysis based on competing-risk regression could bring further strength to the findings.
- The authors correctly discuss the need for external validation. Please consider including an external validation cohort already in this paper to increase the strength of the findings.

Reviewer #3 (Remarks to the Author):

This article examined the effects of pattern of drinking, genetic risk and metabolic factors on the risk of alcohol-related liver disease, as well as their interactions. To enhance the manuscript, I have following comments:

1. The analysis excluded about 10% of participants due to missing covariates. Given the different patient characteristics distributions in those excluded participants, it seems that the missing mechanism is not missing-completely-at-random. The authors could perform some better methods to handle the missing data (such as multiple imputations for survival models) as a sensitivity analysis to examine the robustness of the conclusions.

Reference: White IR, Royston P, Wood AM. Multiple imputation using chained equations: issues and guidance for practice. *Statistics in medicine*. 2011 Feb 20;30(4):377-99.

2. Supplemental Table 1: When there are cell number < 5 (the row for AH), the Fisher's exact test should be used (better than chi-square test).

3. The alcohol drinking pattern is measured at baseline. However, this might change over time (e.g., intervention during the follow-up time), which was not reflected in current models. The authors could indicate this limitation in Discussion.

4. Figure 1-3: It seems that some models include interactions. It would be better to clarify all model terms in figure captions.

5. Figure 4: The label for y-axis ("HR (95%CI)") is incorrect.

6. Section "Statistical Analyses" says "S, the synergy index". However, later the Results section and Figures/Tables used "SI" instead of "S".

7. Supplemental Table 3 &4: sometimes there are two points in the values (e.g., "-0..63").

Response to Reviewers' Comments: NCOMMS-23-02300A - "Binge-Pattern Alcohol Consumption and Genetic Risk Factors as Determinants of Risk of Alcohol-Related Liver Disease"

Reviewer #1:

The significance of this research is high in the field of alcohol-related liver disease as so far there is no comprehensive risk assessment algorithm that can truly identify patients at risk. The robust interaction between genetic risk, and presence of diabetes with pattern of drinking (daily drinking vs binge vs heavy binge) is new information. This study adds to the current knowledge that is very much relevant clinically.

We thank the reviewer for these positive comments.

The manuscript can be improved with addition of further information on the PRS, such as number of SNPs included and perhaps even their list in supplemental data. What is the algorithm the authors used for calculating PRS. What are the cut-offs for low/medium/high PRS and how was it decided? It was difficult to get precise information for this study from the reference provided.

Thank you for this suggestion. To make this easier, we have added the Polygenic Score Catalog ID (PGS000704; searchable at: www.pgscatalog.org) which provides full details of the PRS. The cut-offs for low, medium and high risk was according to quintile, as described in Methods (page 5, paragraph 4).

UK Biobank cohort, although large per se, is control heavy with very few cases especially for AH in this study which may affect the results. Discuss UK Biobank dataset which is control heavy compared to cases (both ARLD and AH) and what implications that might have on the results/interpretation. Drinking information is based on self-report and subject to bias when replicating the study.

The reviewer is correct to note that the proportion of liver disease cases is small. In the discussion, we mention that "our cohort was insufficiently powered for three-way interactions and consequently these analyses must be considered exploratory". Nevertheless, the 870 cases within UK Biobank represent one of the largest ARLD cohorts with detailed alcohol use data; larger than other comparable studies, e.g. Danish Cancer, Diet and Health Study: 342 cases (Askgaard et al., PMID: 25634330).

Regarding the number of alcoholic hepatitis (AH) cases, we agree this number is small. However, AH is occasionally challenging to diagnose without liver biopsy, and consequently may be under-reported in ICD coding data. Additionally, AH was not the primary endpoint for this study. Nevertheless, our data demonstrates a robust association of binge-pattern drinking with risk of AH, which is biologically credible and consistent with expert opinion, yet has not been frequently demonstrated in cohorts. We believe this reinforces the quality of our data; we discuss the AH association in the Discussion (paragraph 3).

Finally, we have added that drinking information is based on self-report within the limitations section of the Discussion (page 12, paragraph 2).

Sharing a proposed risk algorithm for interactions based on their data will be useful for testing in other cohorts for furthering this field.

We appreciate this viewpoint regarding the potential benefits of sharing a proposed risk algorithm for interactions based on our data. However, after careful consideration, we believe that developing an algorithm at this stage would be beyond the scope of the current project, and would require considerable further time and resource which would hinder the dissemination of the current findings.

The aim of our work was to investigate novel associations for the development of severe alcohol-related liver disease (ARLD). Accordingly, our data demonstrates novel interactions of pattern of drinking, genetic factors and diabetes with risk of ARLD. We agree with the reviewer that development of a risk algorithm would be the next step to translate these findings, but this is not a

trivial task and would require additional research, validation, and collaboration with experts in public health modelling to ensure utility and accuracy.

Consequently, we believe that development of a risk algorithm would be a new project, incorporating relevant data from this and complementary studies, and leveraging advancements in data science methodologies.

Results, para 1: The sentence below is not discussed in the manuscript. Can the authors speculate on the high incidence in these patients: “Compared to the final study cohort, the participants with incomplete data had lower weekly alcohol consumption, were less likely to be binge or heavy binge drinkers but had higher ARC and AH incidence.”

It is a common phenomenon that those participants lost to follow-up or with incomplete data have worse health outcome than those who remain in the study (healthy cohort effect). There is also evidence to suggest that previously heavy drinkers who stop drinking or reduce consumption due to poor health, appear to have low alcohol consumption and worse outcome (sick-quitter hypothesis) (Sarich et al., PMID: 30758044). We have added a sentence to address this in the limitations section of the Discussion.

Minor:

Check for typos throughout the manuscript

A bit more definition of ‘monotonic’ in the introduction will be helpful for readers non-conversant with this terminology

Thank you; we have checked the manuscript for typos and added a definition of a monotonic relationship to the introduction.

Reviewer #2:

This is a well-performed analysis from the large UK Biobank dataset. The authors found a significant additive interaction effect between heavy binge drinking and high genetic risk for alcohol-related cirrhosis. A similar additive interaction effect was found between diabetes and heavy binge drinking and between diabetes and high genetic risk for alcohol-related cirrhosis. These findings support the observations from previous studies, which have found considerable interaction effects between various levels and types of alcohol use and metabolic and/or genetic factors for liver disease. The current study is an important piece of work that adds on previous work and extend the analyses in several ways. However, I suggest toning down the statements of novelty (“...demonstrate, for the first time,...” etc).

We agree, and have moderated the text in the Abstract and Discussion to decrease any emphasis on novelty, and focus on the positive findings of the study.

Some issues deserve clarification:

-The mean age was quite high, around 70 years. Were the findings similar in younger individuals? Were there differences between men and women in the findings?

We found no interaction between alcohol consumption group and sex or continuous age in relation to the risk of ARC and AH (p for interaction ranged from 0.169 to 0.490), or between PRS group and sex or continuous age (p for interaction ranged from 0.256 to 0.935). These findings suggest that the effect of binge-pattern drinking and PRS is not significantly different between men and women or across different ages. We have incorporated these findings into the Results section (page 8, paragraph 2, and page 8, paragraph 3).

-There are many approaches to estimating total alcohol intake in the UK Biobank data. Has the approach that the authors used been validated or used previously?

We were particularly interested in pattern of drinking – not just average volume per week. We measured total volume and frequency of consumption using standard techniques (Rehm et al., PMID: 9603301). Subsequently, we integrated the frequency of drinking question with the volume to create categories described in the methods. This derivation of a binge drinking measure from

standard quantity-frequency instruments that use quantities ‘typically’ or ‘usually’ consumed on a given occasion or day has been widely used previously (Foster et al., PMID: 26851816).

-Is there an issue with multicollinearity in the models that seem to include both drinking pattern (binge) and total weekly alcohol intake? How did you assess and treat such collinearity?

We used the variance inflation factor (VIF) to assess the degree of multicollinearity among model covariates including binge-drinking pattern and total weekly alcohol intake. The resulting VIF values in all models were well below 5 (ranging from 1.01 to 2.08), indicating that there was unlikely to be an issue of multicollinearity.

We have included this information in the Methods section (page 6, paragraph 2): “Covariates were checked for multicollinearity using the variance inflation factor (VIF), but no multicollinearity was noted (all VIF \leq 2.08).”

-What about the competing risk of death without liver disease? An additional analysis based on competing-risk regression could bring further strength to the findings.

We agree; as suggested, we repeated all the analyses (except for the 3-way interaction analysis) using competing risk regression models based on the Fine and Gray method¹ to account for the potential competing risk of death from non-ARLD causes. Results of these analyses are presented in Supplemental Tables 5-8 and are described in the Results section (page 10, paragraph 1): “Results and interpretation did not change when we accounted for the potential competing risk of death from non-ARLD causes (Supplemental Tables 5-8) ...”

We are unable to fit a competing risk model for the 3-way interaction due to a zero-cell count, and as such we have used a Firth correction to estimate the coefficient in our main analyses. However, implementing this correction in the context of competing risk regression models in either Stata or R is currently not possible. Nevertheless, the 3-way interaction is unlikely to be affected since the earlier models yielded consistent conclusions between our main analyses and those considering death as a competing risk, as mentioned above.

Reference:

1. Fine, J. P., & Gray, R. J. (1999). A Proportional Hazards Model for the Subdistribution of a Competing Risk. *Journal of the American Statistical Association*, 94(446), 496–509

-The authors correctly discuss the need for external validation. Please consider including an external validation cohort already in this paper to increase the strength of the findings.

We validated our findings in an external cohort, the China Kadoorie Biobank (CKB), with the following considerations:

- Analyses in CKB were restricted to males as only 2% of females reported drinking.
- Approximately 20% of the entire cohort were genotyped in CKB, resulting in over 30,760 males with PRS data after excluding prior liver cirrhosis or hepatitis. Preliminary analysis showed that per one SD increase in PRS was directionally consistent but not significantly associated with ARC (HR 1.13; 95%CI 0.86, 1.48) and AH (HR 1.13; 95%CI 0.65, 1.97) in the genotyped sub-cohort, possibly due to a lack of statistical power (Supplemental Table 18). Therefore, we have chosen not to pursue the PRS based analyses and instead used the maximum available cohort of male weekly drinkers in CKB (N=69,039), which is most consistent with the analyses performed in the UKB.

In the CKB sample, binge-pattern drinking was associated with higher risk of ARC and AH, independent of total weekly alcohol intake (Supplemental Table 15). Within each alcohol group, diabetes conferred a higher risk of ARC (Supplemental Table 16); note that the two lowest consumption groups were combined due to zero cases of ARC in participants who reported drinking below daily limit and had diabetes. Diabetes had additive interaction with heavy binge drinking, with a RERI of 4.37 (95%CI -1.14, 9.88; Supplemental Table 17). These results were in line with our UKB findings.

To further confirm the results from main analyses (including the PRS findings), we also performed a cross-validation by randomly splitting our UKB sample into two independent sub-samples (evenly balanced by age and sex) and then repeated all the analyses in each sub-sample. Results from both sub-samples also supported our initial findings, albeit with wider CIs due to reduced sample sizes (Supplemental Tables 19-23). Although there were some differences between the point HR estimates in the main analyses and in CKB or the UKB sub-samples for certain strata with fewer ARC/AH cases, the CIs often overlapped and included the point estimates.

We have incorporated the above information into the Methods section (page 6, paragraph 3) and the Results section (page 10, paragraphs 2-4).

Reviewer #3:

This article examined the effects of pattern of drinking, genetic risk and metabolic factors on the risk of alcohol-related liver disease, as well as their interactions. To enhance the manuscript, I have following comments:

*1. The analysis excluded about 10% of participants due to missing covariates. Given the different patient characteristics distributions in those excluded participants, it seems that the missing mechanism is not missing-completely-at-random. The authors could perform some better methods to handle the missing data (such as multiple imputations for survival models) as a sensitivity analysis to examine the robustness of the conclusions. Reference: White IR, Royston P, Wood AM. Multiple imputation using chained equations: issues and guidance for practice. *Statistics in medicine*. 2011 Feb 20;30(4):377-99.*

We agree that it is important to assess the impact of missing data on our findings. In response to this concern, we implemented multiple imputation by chained equations¹ as a sensitivity analysis. ARLD was represented by an indicator variable, and the Nelson-Aalen cumulative hazard estimator was utilised.² A total of 10 imputations were generated; the number of imputations should be at least equal to the percentage of incomplete cases, which was 8.7% in our study. We have incorporated these details into the Methods section (page 6, paragraph 3) and cited the two references.

Results based on imputed data are presented in Supplemental Tables 9-13 and are described in the Results section (page 10, paragraph 1): “Results and interpretation did not change when we used multiply imputed datasets (Supplemental Tables 9-13).”

References:

1. White IR, Royston P, Wood AM. Multiple imputation using chained equations: issues and guidance for practice. *Statistics in medicine*. 2011 Feb 20;30(4):377-99
2. White IR, Royston P. Imputing missing covariate values for the Cox model. *Stat Med*. 2009 Jul 10;28(15):1982-98

2. Supplemental Table 1: When there are cell number<5 (the row for AH), the Fisher's exact test should be used (better than chi-square test).

We have performed the Fisher's exact test for the row of AH, and updated the footnote accordingly: “**** P values from one-way ANOVA or chi-square test (Fisher's exact test for cells with <5 observations) to compare the differences across binge groups.”

3. The alcohol drinking pattern is measured at baseline. However, this might change over time (e.g., intervention during the follow-up time), which was not reflected in current models. The authors could indicate this limitation in Discussion.

We agree; a statement has been added in the Discussion section to acknowledge this limitation (page 12, paragraph 2): “Additionally, data regarding pattern of drinking was only collected at baseline.”

4. Figure 1-3: It seems that some models include interactions. It would be better to clarify all model terms in figure captions.

The captions for Figures 1-3 have been revised to clarify all the terms in each model, including the interactions:

Figure 1: “HRs were derived from Cox model with following covariates: alcohol consumption group, sex, age, ethnicity, Townsend deprivation index at recruitment, physical activity, smoking, total weekly alcohol intake, beverage type, drinking with/without meal, prevalent diabetes and BMI categories.”

Figure 2: “HRs were derived from Cox model with following covariates: interaction of alcohol consumption and PRS groups, total weekly alcohol intake, diabetes, BMI, sex, age, ethnicity, Townsend deprivation index, physical activity, smoke, alcohol type, drinking with/without meals, genotyping array, first 10 ancestry principal components.”

Figure 3: “HRs were derived from Cox models with following covariates: sex, age, ethnicity, Townsend deprivation index at recruitment, physical activity, smoking, total weekly alcohol intake, beverage type, drinking with/without meal, BMI, genotyping array, first 10 ancestry principal components plus PRS group, interaction of alcohol consumption group and diabetes for (a), and alcohol consumption group, interaction of PRS group and diabetes for (b).”

5. *Figure 4: The label for y-axis (“HR (95%CI)”) is incorrect.*

The label has been corrected to ‘HR’. Corresponding changes have been made to the caption, which now reads as follows:

“Figure 4: Hazard ratios for risk of alcohol-related cirrhosis with contributions of heavy binge, high PRS, diabetes and their combination (interaction). Baseline represents the risk in referent participants who reported drinking within daily limits, had low PRS and were free of diabetes.”

6. *Section “Statistical Analyses” says “S, the synergy index”. However, later the Results section and Figures/Tables used “SI” instead of “S”.*

We have made the necessary changes in the Results section and Figures/Tables (Supplemental Tables 3 and 4) to ensure consistency with the rest of the manuscript by using the same abbreviation, ‘S’, to refer to the synergy index.

7. *Supplemental Table 3 &4: sometimes there are two points in the values (e.g., “-0..63”).*

We have carefully reviewed the tables and removed the instances where there were two points in the values.

REVIEWERS' COMMENTS

Reviewer #1 (Remarks to the Author):

The authors have addressed concerns to my satisfaction in the revised manuscript and response to reviewers.

Reviewer #2 (Remarks to the Author):

The authors have sufficiently addressed my previous concerns

Reviewer #3 (Remarks to the Author):

The authors have addressed my comments adequately. No further comments.